# Heterologous Expression of Secondary Metabolite Genes in *Trichoderma reesei* for Waste Valorization

**DOI:** 10.3390/jof8040355

**Published:** 2022-03-30

**Authors:** Mary L. Shenouda, Maria Ambilika, Elizabeth Skellam, Russell J. Cox

**Affiliations:** 1Institute for Organic Chemistry and Biomolekulares Wirkstoffzentrum (BMWZ), Schneiderberg 38, 30167 Hannover, Germany; mary.shenouda@oci.uni-hannover.de (M.L.S.); m.ambilika@gmail.com (M.A.); elizabeth.skellam@unt.edu (E.S.); 2Pharmacognosy Department, Faculty of Pharmacy, Alexandria University, Alexandria 21521, Egypt; 3Department of Chemistry and BioDiscovery Institute, University of North Texas, 1155 Union Circle, Denton, TX 76201, USA

**Keywords:** heterologous expression, PKS-NRPS, PKS, waste valorization, microbial cell factory, *Trichoderma reesei*

## Abstract

*Trichoderma reesei* (*Hypocrea jecorina*) was developed as a microbial cell factory for the heterologous expression of fungal secondary metabolites. This was achieved by inactivation of sorbicillinoid biosynthesis and construction of vectors for the rapid cloning and expression of heterologous fungal biosynthetic genes. Two types of megasynth(et)ases were used to test the strain and vectors, namely a non-reducing polyketide synthase (nr-PKS, *aspks1*) from *Acremonium strictum* and a hybrid highly-reducing PKS non-ribosomal peptide synthetase (hr-PKS-NRPS, *tenS + tenC*) from *Beauveria bassiana*. The resulting engineered *T. reesei* strains were able to produce the expected natural products 3-methylorcinaldehyde and pretenellin A on waste materials including potato, orange, banana and kiwi peels and barley straw. Developing *T. reesei* as a heterologous host for secondary metabolite production represents a new method for waste valorization by the direct conversion of waste biomass into secondary metabolites.

## 1. Introduction

Heterologous expression of fungal biosynthetic gene clusters (BGC) is an effective method for the synthesis of known, and engineering of new, natural products [1,2,3]. Heterologous hosts include *Escherichia coli* [4], *Saccharomyces cerevisiae* [5], *Aspergillus oryzae* [1] and *Aspergillus nidulans* [6]. Many common host organisms such as *S. cerevisiae* and *E. coli* are conveniently manipulated, but require tight control of fermentation conditions and media components [1,5,7,8,9,10,11]. However, filamentous fungi offer high potential as hosts for secondary metabolite production as they can grow rapidly on a wide variety of substrates. In particular, *Trichoderma reesei* (*Hypocrea jecorina*) is a filamentous fungus well known for its industrial production of cellulases and other cell wall-degrading enzymes [12]. *T. reesei* has long been used as a heterologous host for enzyme production due to its excellent protein production capabilities and its Generally Recognized as Safe (GRAS) status [13]. However, the use of *T. reesei* as a heterologous host for secondary metabolite production has not been reported so far.

The ability of *T. reesei* to produce high levels of cellulases has exploited the organism’s ability to grow to high cell mass on low-value materials such as waste streams from agriculture, food production and paper and cardboard production [14,15,16]. This contrasts with other fungi used in biotechnology such a *A. oryzae* or *A. nidulans* that usually require high-quality or food-grade substrates. The use of *T. reesei* in waste valorization by converting cellulosic biomass to fuels and chemicals was pioneered by Reese and Mandels after the discovery of *T. reesei* cellulases that degraded the tent canvas of the US army during World War II [17]. Since then, enhancing the production levels of cellulases from *T. reesei* has been a significant focus of research [18]. We reasoned that *T. reesei* should also make an effective host for heterologous secondary metabolite production and the aim of this work is to develop *T. reesei* as a microbial cell factory for secondary metabolite production by direct fermentation of different types of waste materials.

## 2. Materials and Methods

All chemicals and media ingredients used in this work were purchased from Duchefa Biochemie (Haarlem), Roth (Karlsruhe), VWR (Darmstadt), Fisher scientific (Schwerte), Sigma Aldrich (Taurkirchen), abcr (Kahlsruhe) and Formedium (Hunstanton). Double-distilled water (dd H_2_O) was used for the preparation of all media, buffers, solutions, and antibiotics. Sterilization of all growth media and solutions was achieved by autoclaving at 121 °C for 15 min using Systec VX150 or Prestige Medical 2100 autoclaves. Sterilization of antibiotic solutions was achieved using a sterile syringe filter (0.45 µm pore size, Roth). For details of all growth media, buffers and solutions, enzymes, and antibiotics, see electronic Appendix A.

### 2.1. Microbiological Methods

Bacterial and fungal strains used in this work are summarized in the Appendix A. All the microbiological methods used were the same as those published previously [10].

### 2.2. Molecular Biology Methods

All enzymes were purchased from NEB and Thermo Fisher Scientific and were used according to the manufacturer’s protocols using the supplied buffers. All the vectors and oligonucleotides used in this work are summarized in the Appendix A. All molecular biology methods used were the same as published previously [10].

### 2.3. Fermentation

For potato, orange, banana and kiwi peels, 50 g of the peels were cut into small pieces and autoclaved with 100 mL pure water at 121 °C for 15 min. For coffee press, 10 g coffee press left-over from coffee machine was autoclaved (121 °C, 15 min) with 100 mL pure water. For barley straw, 5 g barley straw (donated from a horse stable near Hannover, Germany) was cut into small pieces and autoclaved with 100 mL pure water at 121 °C for 15 min. *Trichoderma reesei* transformants were grown on minimal media agar plates for 3–5 days. Mycelia and spores were scraped off using a sterile spatula. Finally, 250 µL of this suspension were inoculated on the waste media. The flasks were incubated for 14 days at 28 °C without shaking.

### 2.4. Extraction of Transformants Grown on Waste Materials

For potato, orange, banana, and kiwi peels as well as coffee press, the fungal mycelia together with the media components were homogenized at the end of the fermentation period using a hand blender. This was followed by filtration and the filtrate was acidified with 2 M HCl before extraction twice with ethyl acetate. The ethyl acetate fraction was then dried over anhydrous MgSO_4_ and evaporated under reduced pressure. The residue was dissolved in acetonitrile to reach a final concentration of 5 mg/mL.

For barley straw, 200 mL of ethyl acetate was added to the flasks containing the fungal mycelia and the straw and stirred using a magnetic stirrer for 4 h before filtration. The ethyl acetate fraction was then separated, dried over anhydrous MgSO_4,_ and evaporated under reduced pressure. The residue was dissolved in acetonitrile to reach a final concentration of 5 mg/mL.

### 2.5. Chemical Analysis

All the chemicals and materials were purchased from one of the following companies: Bio-Rad (München, Germany), New England Biolabs (Beverly, MA, USA), Roth (Karlsruhe, Germany), Sigma Aldrich (Steinheim, Germany), and Thermo Fisher Scientific (Waltham, MA, USA). All chemical analysis methods used were the same as previously described [10].

## 3. Results

### 3.1. Construction of a Host with a Cleaner Secondary Metabolic Background

*T. reesei* is a prolific producer of sorbicillinoids, derived from sorbicillin **1** and sorbicillinol **2** (Figure 1). The sorbicillin pathway is well understood and the encoding sorbicillin (*sor*) BGC has been investigated in detail (Figure 1A) [19,20,21]. Sorbicillinoid compounds make up the majority of the natural products produced in *T. reesei* fermentations and include previously identified compounds such as epoxysorbicillinol **3**, bisorbicillinol **4** and bisvertinolone **5**. We decided to prevent the biosynthesis of sorbicillinoids to give a cleaner secondary metabolic background and potentially enable a higher proportion of metabolic precursors to be used for the production of desired metabolites. Therefore, we aimed to knock out key *sor* biosynthetic genes using the classical bipartite method of Nielsen that dramatically decreases the chances of false-positive transformants [22].

A vector was constructed using yeast homologous recombination to knock out (KO) the adjacent *sorB* and *sorC* genes simultaneously. The vector was constructed using the pEYA backbone [23] and a hygromycin resistance gene (*hph* [24]) inserted between *P_gpdA_* and *T_trpC_* with ≈1 kb regions homologous to the 5’ end of *sorB* and 3’ end of *sorC* (see Appendix A). This vector was used as a template for the construction of overlapping PCR fragments for the bipartite KO of the *sorB/sorC* region. *T. reesei* QM6a·Δ*tmus53·*Δ*pyr4* (gift of Prof. Dr. Astrid Mach-Aigner, TU Wien [25]) was transformed with these PCR fragments. After three rounds of selection on PDA containing hygromycin, eighteen hygromycin-resistant transformants were selected. Chemical analysis of six of them showed that one transformant, *T. reesei* QM6a·Δ*tmus53*·Δ*pyr4*·Δ*sorBC*, showed no production of any sorbicillin-related compounds after cultivation for seven days in DPY + 1% glucose (LC–MS analysis, Figure 2). Cultivation of this transformant on different media revealed its inability to produce sorbicillinoids under all tested conditions. PCR analysis and partial sequencing confirmed the expected loss of the *sorB* and *sorC* genes.

Although the *T. reesei* QM6a·Δ*tmus53*·Δ*pyr4*·Δ*sorBC* strain did not produce any sorbicillin-related compounds, it was able to produce scytolide **6** as previously observed [20]. In addition, new compounds were produced that were not previously observed in *T. reesei* QM6a·Δ*tmus53·*Δ*pyr4*. One of these compounds was present in a concentration high enough to allow its isolation and structure elucidation using NMR and comparison to published data (See Appendix A). The compound was identified as the known citreoisocoumarin **7** (Figure 1B) [26].

### 3.2. Construction of a T. reesei Heterologous Expression Vector

In our previous work, we have made extensive use of the modular vector system developed by Lazarus and coworkers for use in *Aspergillus oryzae* [27,28]. Due to the low availability of vectors for gene integration in *T. reesei,* an *A. oryzae* vector was adapted as the backbone to construct a new *T. reesei* expression system. The vector was constructed based on the pTYGS·*argB* vector [29], where the inducible *Aspergillus oryzae amyB* promoter *(P_amyB_)* was replaced with the pyruvate decarboxylase promoter (*P_pdc_*) from *T. reesei* itself and the *Aspergillus argB* selection marker was replaced with the *pyr4* marker using homologous recombination in yeast (Figure 3). These changes were designed to allow easy and high-throughput heterologous expression of megasynth(et)ases in *T. reesei*.

The constructed vector, pTYGS·*pyr4*·*P_pdc_*, has *att*R sites that allow in vitro recombination of a target gene present on a Gateway entry vector, downstream of *P_pdc_*. In an initial experiment, we wished to use a reliable and well-understood synthase to test the system. We therefore selected *aspks1* that encodes the non-reducing polyketide synthase (nr-PKS) methylorcinaldehyde synthase (MOS) from the xenovulene biosynthetic pathway of *Acremonium strictum* [30]. The entry vector pEYA·*aspks1* [27] was recombined in vitro with pTYGS·*pyr4*·*P_pdc_* to insert *aspks1* into the cloning site downstream of *P_pdc_*. The resulting vector, pTYGS·*pyr4*·*P_pdc_*·*aspks1*, was confirmed by PCR and sequencing.

The vector pTYGS·*pyr4*·*P_pdc_*·*aspks1* was transformed into *T. reesei* QM6a·∆*tmus53*·∆*pyr4*·∆*sorBC* using standard PEG-mediated transformation (See ESI) [31]. After three rounds of selection on minimal media lacking uridine, eight transformants were obtained and cultivated on PDB media for 48 h at 28 °C and 110 rpm. Extraction of all of the transformants showed that seven out of the eight transformants produce the expected natural product 3-methylorcinaldehyde **8** in addition to minor amounts of the corresponding 3-methylorsellinic acid **9** (Figure 2C). Longer cultivation periods and fermentation on different media showed an increase in the production levels of the acid **9** and a decrease in the aldehyde **8** production (see Appendix A). Analysis of the gDNA of all eight transformants showed the correct insertion of *aspks1* in seven out of the eight transformants (see Appendix A), in agreement with the LC–MS analysis.

### 3.3. Growing Recombinant T. reesei Strains on Different Waste Materials

The ability of the new transformant, *T. reesei* QM6a·∆*tmus53*·∆*sorBC*·*P_pdc_*·*aspks1*, to grow on different waste materials and produce **8** and **9** was tested. Different substrates were used such as potato peel, orange peel, banana peel, kiwi peel, coffee grinds and barley straw. The fungus grew quickly on all substrates, except orange peels. Fermentations were extracted after 14 days of cultivation with ethyl acetate.

LC–MS analysis of the extracts showed the ability of the transformed strain to produce methylorcinaldehyde **8** and methyl orsellinic acid **9** on four out of the five tested media but in different ratios (Figure 4). Fermentation on coffee grinds did not appear to produce either of **8** or **9**. Quantification of the production of these compounds on potato peels showed that the strain was able to produce up to 371 mg·kg^−1^ dry weight of combined **8** and **9** (See Appendix A) without further optimization. The orange peel fermentation grew very slowly and was incubated at 28 °C for four months. Extraction of this culture showed the production of 3-methylorcinaldehyde **8**, albeit at low titer (see Appendix A).

### 3.4. Construction of a Multiple-Gene Expression System

We were next interested to test the ability of *T. reesei* to express more than one biosynthetic gene in parallel. Additional native promoters were therefore chosen to be inserted in the pTYGS vector. The strongest known constitutive promoters in *T. reesei* are those driving expression of cDNA1 (*P_cDNA1_*) and enolase (*P_TReno_*) [32]. Therefore, the two promoters, *P_cDNA1_* and *P_TReno_* were chosen to expand the vector pTYGS·*pyr4·P_pdc_*.

*P_cDNA1_* and *P_TReno_* coding regions were amplified from *T. reesei* gDNA using primers with overhangs homologous to the pTYGS·*pyr4*·*P_pdc_* vector backbone and to the terminators *T_adh_* and *T_eno_*, respectively. Inclusion of a *Swa*I restriction site downstream of *P_cDNA1_* and *P_TReno_* facilitated later gene insertion by yeast homologous recombination. The vector was then linearized using *Asc*I and the new vector was constructed by yeast homologous recombination between the linearized vector pTYGS·*pyr4*·*P_pdc_*, the coding sequence of *P_cDNA1_* and *P_TReno_* and PCR-derived patches to repair unused *Asc*I restriction sites. The resultant vector was confirmed by PCR and sequencing and named pTYGS·*pyr4*·*P_pdc_*·*P_cDNA1_*·*P_TReno_* (Figure 5).

To test the ability of the pTYGS·*pyr4*·*P_pdc_*·*P_cDNA1_*·*P_TReno_* system, a two-gene biosynthetic pathway, was selected. The well-studied tenellin BGC was chosen as it requires the cooperation of a PKS-NRPS encoded by *tenS* with a *trans*-acting enoyl reductase encoded by *tenC* [33]. Therefore, a new vector (pTYGS·*pyr4*·*P_pdc_*·*tenS*·*P_cDNA1_*·*tenC*·*P_TReno_*) was constructed with *tenS* under the control of *P_pdc_* and *tenC* driven by *P_cDNA1_* (Figure 5). Transformation of *T. reesei* QM6a ∆*tmus53*·∆*pyr4*·∆*sorBC* with pTYGS·*pyr4*·*P_pdc2_*·*tenS*·*P_cDNA1_*·*tenC*·*P_TReno_* resulted in the production of nine transformants. The transformants were selected three times on minimal media and finally transferred onto PDA plates. Five out of the nine transformants were cultivated in DPY + 1% glucose for three days followed by extraction with EtOAc and LC–MS analysis. This showed the production of the expected natural product pretenellin A **10** in all the tested transformants (Figure 6A) [34].

PCR analysis of the gDNA of four different *T. reesei* QM6a·∆*tmus53*·∆*sorBC*·*P_pdc_*·*tenS*·*P_cDNA1_*·*tenC* transformants showed the correct insertion of *tenS* and *tenC* in all the tested transformants (Appendix A). The best-producing transformant was then cultivated on autoclaved banana peels to test the ability of the transformed strain to grow on waste materials and produce the expected compound. The LC–MS chromatogram of the transformant after 12 days of growth showed the production of the expected compound **10** (Figure 6B).

## 4. Discussion

Due to its impressive ability to produce high amounts of cellulases, *T. reesei* has been developed as an effective heterologous host for protein production. Its safety and high production capacity are very appealing and have led to increased attention on *T. reesei* in recent decades, especially after the publication of its full genome sequence in 2008 [35]. Therefore, many toolkits for transforming *T. reesei* and for the expression of proteins and reporter genes have been developed [24,36,37]. These include the development of different auxotrophic strains to facilitate the transformation, identification of many native promoters to increase and control protein expression and different methods for transformation [19].

Despite this great interest in *T. reesei* as a heterologous host for protein production, almost no research has been performed on *T. reesei* as a heterologous host for secondary metabolite production. We aimed to test the suitability of *T. reesei* as a heterologous host for the expression of fungal genes to produce two model natural products. *T. reesei* is evidently able to effectively produce secondary metabolites, being a well-known producer of the sorbicillinoids [19,20,38,39,40]. Several other secondary metabolites are known including the polyketide trichodermatides A-D [41], and the cyclic tetrapeptide trichoderide A [42], and we have recently demonstrated that the organism harbours a fully functional ilicicolin H BCG that is silent under laboratory conditions [10].

The production of sorbicillinoids is known to hamper the production of secondary metabolites in industrial filamentous fungi. Sorbicillin production was previously eliminated in the beta-lactam producer *Penicillium chrysogenum* [43] as part of classical strain-improvement strategies. Knock out of the *T. reesei* sorbicillin BGC would result in several advantages. First, sorbicillin production likely consumes available acetyl and malonyl-CoA and cofactors that could be better directed towards production of desired metabolites. Secondly, the high levels of sorbicillinoids in the extract complicate the isolation of desired metabolites. Finally, recent studies have shown negative effects of sorbicillinoid production on growth, conidiation, cell wall integrity and cellulase production in *T. reesei* [2,44]. Therefore, a *T. reesei* strain lacking sorbicillin synthesis was desired. Previous attempts to knock out this gene cluster from *T. reesei* have been successful but they were achieved in the wild-type strain [19,20]. However, we required a strain possessing auxotrophic selection markers to allow the later selection of expressed biosynthetic genes.

Bipartite knock out of the *sorB*/*sorC* region was easily achieved and the resulting strain *T. reesei* QM6a·∆*tmus53*·∆*pyr4*·∆*sorBC* strain could not produce sorbicillinoids under any tested conditions. However, some new compounds, including citreoisocoumarin **7**, were produced in observable amounts in this strain, possibly due to the increased availability of acetyl-CoA and other building blocks. Although this compound was not previously reported from *T. reesei*, it has been isolated from the sponge derived *Trichoderma* HPQJ-34 and is a common natural product isolated from other fungi and is likely to be a shunt from the alternariol biosynthetic pathway [45]. Bioinformatic analysis of all PKS genes from *T. reesei* [10] showed the presence of a gene encoding a PKS with high similarity (67%) to the nr-PKS (PkgA) from *A. nidulans*, which was reported to produce citreoisocoumarin and related compounds [46] and the PKS is closely related to snPKS19 from *Parastagonospora nodorum* that is known to synthesize alternariol [47].

The vector pTYGS·*pyr4*·*P_pdc_* was then constructed from the well-known *Aspergillus* expression vector pTYGS·*argB*. It has all the advantages of the pTYGS vectors including the Gateway^®^ in vitro recombination system and the ability to shuttle between yeast and *E. coli* to facilitate gene insertion by yeast homologous recombination. The *aspks1* gene was then used to test this system. *T. reesei* QM6a·∆*tmus53*·∆*pyr4*·∆*sorBC* strains that were transformed with pTYGS·*pyr4*·*P_pdc_*·*aspks1* were able to produce the expected compounds 3-methylorcinaldehyde **8** and 3-methylorsellinic acid **9** that is presumably derived by facile oxidation of the aldehyde. The same compounds were observed when *aspks1* was expressed in *A. oryzae* [29]. As expected, the new *T. reesei* transformant expressing *aspks1* was able to produce 3-methylorcinaldehyde **8** on different waste materials such as potato, banana, kiwi and orange peels and barley straw. However, the strain showed delayed growth on autoclaved fresh orange peels that might be attributed to the reported antifungal activity of essential oils of orange peels [48,49].

To expand the new system, additional native promoters of *T. reesei* were added to the vector using yeast homologous recombination. The new vector (pTYGS·*pyr4*·*P_pdc_*·*P_cDNA1_*·*P_TReno_*) contains three native constitutive promoters (*P_TReno_*, *P_cDNA1_* and *P_pdc_*) and one *A. nidulans* constitutive promoter (*P_gpdA_*). Constitutive promoters were used instead of potentially stronger inducible promoters involved in cellulase expression to allow for constitutive production of the produced compounds under different cultivation conditions. The constructed vector was used to express the megasynthetase PKS-NRPS gene (*tenS*) together with its *trans*-acting ER (*tenC*) in *T. reesei* by adding the *tenS* gene under control of *P_pdc_* and *tenC* gene under the control of *P_cDNA1_*. *T. reesei* transformants containing this construct were able to produce the expected pretenellin A **10** on different media and on banana peels.

## 5. Conclusions

We have demonstrated that *T. reesei* can be engineered to produce fungal natural products using an adaptation of the highly successful *A. oryzae* pTY expression system developed by Lazarus and coworkers [1,28]. Simple exchange of selection markers and promoters led to expression vectors functional in *T. reesei*. Entry vectors previously constructed for use in the *A. oryzae* system, or rapid recombination in yeast, can easily supply genes that are functionally expressed in *T. reesei.* Since *T. reesei* is easily grown on a range of waste materials, this research paves the way for use of *T. reesei* as a microbial cell factory for waste valorization. The ability of *T. reesei* as a host to grow on low-value and waste substrates offers an important advantage over host organisms such as *E. coli* and *S. cerevisiae*. Production of high-value secondary metabolites in *T. reesei* represents a promising, low-cost, fast, sustainable, and green alternative for synthetic chemistry in the production of secondary metabolites. Removal of sorbicillin metabolites from the expression host makes both analysis by chromatography and chromatographic purification of desired products simpler.

In this research, waste materials such as potato peel, kiwi and banana peel were used as a cultivation media for the producing strain without any pre-treatment. However, further experiments are required to enhance the system such as testing different pre-treatment methods to allow better and faster production of the expected natural products on waste. The promoters used in this work were constitutive promoters to allow the production of natural products under all cultivation conditions, and it may be that inducible promoters previously successfully used for cellulase production could be effectively exploited [35]. Therefore, future experiments would be further enhancement of the system by the construction of new vectors with stronger promoters such as the tunable cellulase promoters *cbh1* and *cbh2* [35]. These promoters could also allow the system to produce a higher concentration of the expected compounds on lignocellulosic waste and other biomass [50].

So far, we have shown that one-gene (e.g., *aspks1*) and two-gene (e.g., *tenS* + *tenC*) biosynthetic pathways can be expressed successfully in *T. reesei*. In the future, expansion of the numbers of genes expressed in parallel should enable more complex pathways to be investigated. Impressive work by Abe and coworkers in *A. oryzae* [51] has shown that up to 12 biosynthetic genes can be co-expressed in parallel to produce highly complex meroterpenoid toxins. It seems feasible that *T. reesei* could also be used to produce such complex metabolites directly from waste materials. Future work will inevitably explore an expansion of the preliminary results demonstrated here.

## Figures and Tables

**Figure 1 jof-08-00355-f001:**
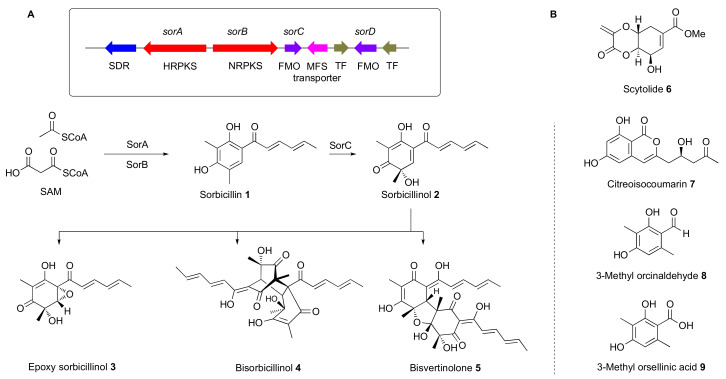
(**A**), the sorbicillin (*sor*) biosynthetic gene cluster in *T. reesei*, and known pathway to monomeric (**1–3**) and dimeric (**4–5**) sorbicillinoids; (**B**), compounds isolated from *T. reesei* QM6a·Δ*tmus53*·Δ*pyr4*·Δ*sorBC* and *T. reesei* QM6a·∆*tmus53*·∆*sorBC*·*P_pdc_*·*aspks1*.

**Figure 2 jof-08-00355-f002:**
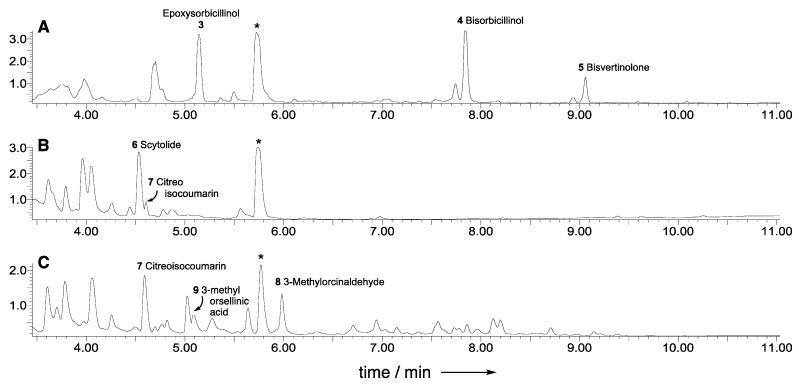
LC–MS traces of *T. reesei* strains grown in on DPY + 1% glucose media. (**A**), *T. reesei* QM6a·Δ*tmus53·*Δ*pyr4*; (**B**), *T. reesei* QM6a·Δ*tmus53*·Δ*pyr4*·Δ*sorBC*; (**C**), *T. reesei* QM6a·∆*tmus53*·∆*sorBC*·*P_pdc_*·*aspks1*. * = unrelated compound.

**Figure 3 jof-08-00355-f003:**
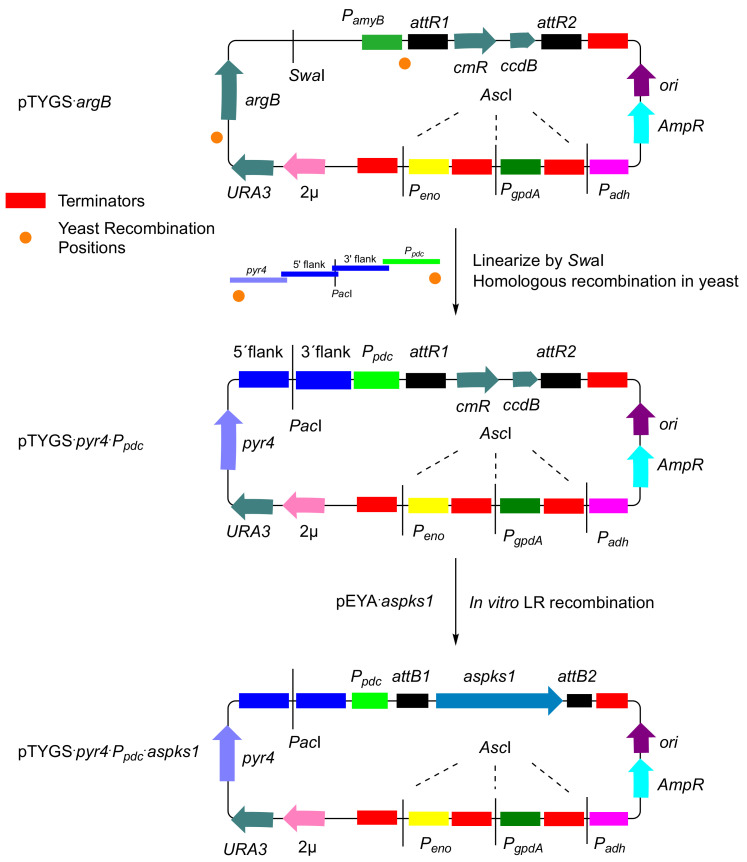
Construction of general expression vector pTYGS·*pyr4·P_pdc_* and nr-PKS expression system pTYGS·*pyr4·P_pdc_·aspks1*.

**Figure 4 jof-08-00355-f004:**
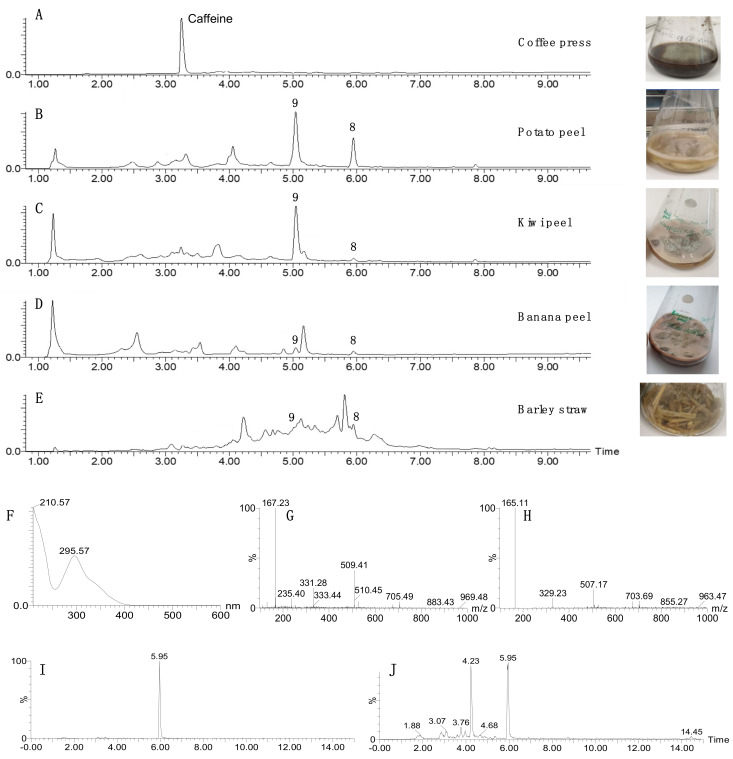
LC–MS traces of *T. reesei* QM6a·∆*tmus53*·∆*sorBC*·*P_pdc_*·*aspks1* cultivated on different media after 14 days of growth. (**A**), grown on coffee press; (**B**), grown on potato peel; (**C**), grown on kiwi peel; (**D**), grown on banana peel; (**E**), grown on barley straw; (**F**), uv diode array detector (DAD) trace of 5.95 min peak; (**G**), ES+ spectrum of 5.95 min peak; (**H**), ES- spectrum of 5.95 min peak; (**I**), extracted ion chromatogram (EIC, ES- 165) for barley straw fermentation; (**J**), EIC chromatogram (ES+ 167) for barley straw fermentation.

**Figure 5 jof-08-00355-f005:**
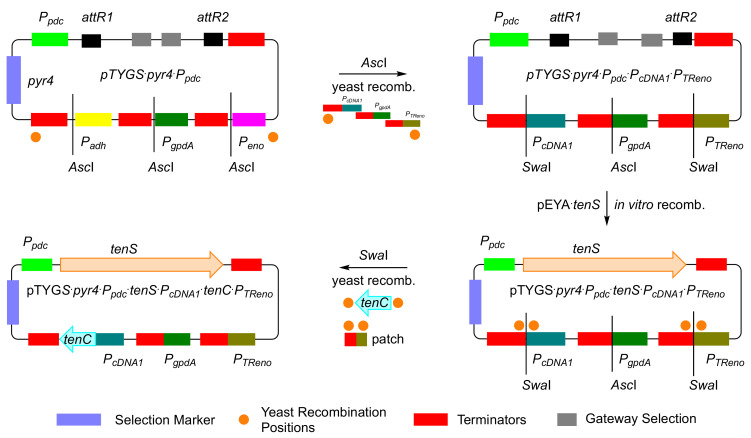
Construction of pTYGS·*pyr4*·*P_pdc_*·*P_cDNA1_*·*P_Treno_* and pTYGS·*pyr4*·*P_pdc2_*·*tenS*·*P_cDNA1_*·*tenC*·*P_Treno_*.

**Figure 6 jof-08-00355-f006:**
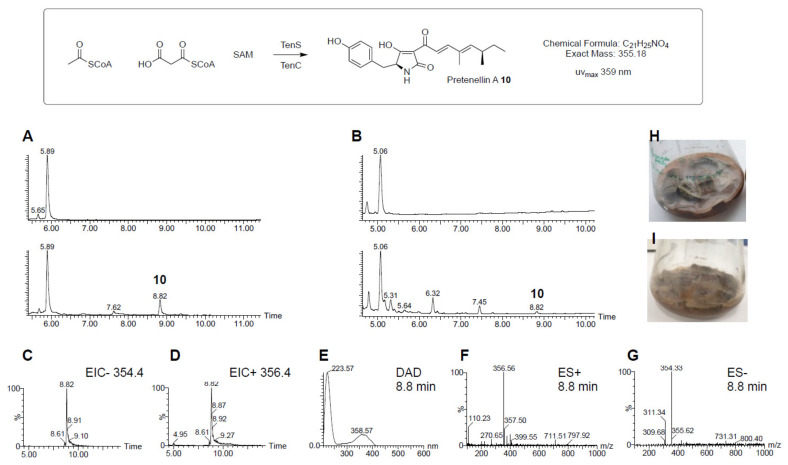
Production of pretenellin A **10**. (**A**), LC–MS traces of *T. reesei* QM6a·∆*tmus53*·∆*sorBC*·*P_pdc_*·*tenS*·*P_cDNA1_*·*tenC* transformant (lower) compared to *T. reesei* QM6a·∆*tmus53*·Δ*pyr4 ∆sorBC* strain (upper) on DPY+ 1% glucose after 3 days of cultivation; (**B**), DAD chromatogram of transformant *T. reesei* QM6a·∆*tmus53*·∆*sorBC*·*P_pdc_*·*tenS*·*P_cDNA1_*·*tenC* on banana peels (lower) in comparison to *T. reesei* QM6a·∆*tmus53*·Δ*pyr4 ∆sorBC* strain (upper); (**C**,**D**), extracted ion chromatograms (EIC) for the expected masses of pretenellin A **10** in the extract of *T. reesei* QM6a·∆*tmus53*·∆*sorBC*·*P_pdc_*·*tenS*·*P_cDNA1_*·*tenC* transformant on DPY + 1% glucose; (**E**), diode array detector (DAD) data for 8.8 min peak; (**F**), ES+ spectrum for 8.8 min peak; (**G**), ES- spectrum for 8.8 min peak; (**H**), growth of *T. reesei* QM6a·∆*tmus53*·∆*sorBC* on banana peel; (**I**), growth of *T. reesei* QM6a·∆*tmus53*·∆*sorBC*·*P_pdc_*·*tenS*·*P_cDNA1_*·*tenC* on banana peel.

## Data Availability

Not applicable.

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
