# Peer review of "Heterologous Expression of Secondary Metabolite Genes in Trichoderma reesei for Waste Valorization"

_jof, 2022, doi:10.3390/jof8040355_

Round 1

Reviewer 1 Report

Heterologous expression has been widely used in the activation of silent gene clusters of natural products, which mainly depends on an effective host and easy fermentation conditions. In this manuscript, Cox and coworkers selected fungus Trichoderma reesei as the target, knocked out the sorbicillinoid biosynthetic genes and constructed an engineered strain, which features the cleaner secondary metabolic background. The authors introduced a nrPKS gene (aspks1) or two hrPKS-NRPS genes (tenS + tenC) into the strain, and the expected natural products were successfully produced on waste materials including potato, orange, banana and kiwi peels and barley straw. I think this study presents a nice example that conversion of waste biomass into fungal compounds and develops T. reesei as a new fungal host for fungal secondary metabolite production. Therefore, I support its publication in JoF, I think it will be of great interest to the broad readership of JoF.

Several minor comments that should be addressed prior to the acceptance of this manuscript.

  1. Figure 2C, T. reesei QM6a·âˆ†tmus53·âˆ†sorBC·Ppdc·aspks1 produced compounds 8 and 9, however, the yield of 7 is increased and compound 6 is disappeared. Could you give some explanations?

  1. The 1D and 2D NMR data of compound 8 and 10 should be provided. The purity of compound 9 should be improved, the 13C-NMR spectrum of compound 9 is not good.

  1. Figure 6B, the production of compound 10 is low, what are the other compounds (tR=5.31 min, tR=6.32 min and tR=7.45 min)?

Author Response

  1. Figure 2C, T. reesei QM6a·âˆ†tmus53·âˆ†sorBC·Ppdc·aspks1 produced compounds 8 and 9, however, the yield of 7 is increased and compound 6 is disappeared. Could you give some explanations?

The data here is qualitative. The data shows the stepwise introduction of genetic elements (A, the starting situation; B, the knockout of the competing sorbicillins; and C, the introduction of biosynthetic genes). Quantitative studies were not performed here as the aim was simply to show that the KO strain would produce 8 and 9. Later work quantifies these more thoroughly.

  1. The 1D and 2D NMR data of compound 8 and 10 should be provided. The purity of compound should be improved, the 13C-NMR spectrum of compound is not good.

These are all known compounds that have been fully characterised in the literature before. The identity of 8, 9 and 10 was shown by chromatography (LCMS) and comparison with standard materials. NMR of 9 is not perfect but proves its identity - further purification will not change this.

  1. Figure 6B, the production of compound 10 is low, what are the other compounds (tR=5.31 min, tR=6.32 min and tR=7.45 min)?

The 7.4 min peak has a mass of 208, and does not contain nitrogen and so is unrelated to 10.

The 6.3 min peak has a mass of 194, and does not contain nitrogen and so is unrelated to 10.

The 5.3 min peak has a mass of 426, , and does not contain nitrogen and so is unrelated to 10.

I have added extra data to the ESI to show this. Future work will of course include titre optimisation - the present work is a proof of concept that waste material can be converted directly to secondary metabolites.

Reviewer 2 Report

This is an excellent manuscript that clearly presents a method for producing valuable natural products via heterologous expression in Trichoderma on food waste.    My one suggestion is in the title -  seems like it is heterologous expression of genes, not heterologous expression of metabolites.    

Author Response

Thanks for the suggestion - I changed the title as suggested.